# PEtab—Interoperable specification of parameter estimation problems in systems biology

**Leonard Schmiester**[1,2☾], **Yannik Schälte**[1,2☾], **Frank T. Bergmann**[3], **Tacio Camba**[4,5], **Erika Dudkin**[6], **Janine Egert**[7,8], **Fabian Fröhlich**[9], **Lara Fuhrmann**[6], **Adrian L. Hauber**[8,10], **Svenja Kemmer**[8,10], **Polina Lakrisenko**[1,2], **Carolin Loos**[1,2,11,12], **Simon Merkt**[6], **Wolfgang Müller**[13], **Dilan Pathirana**[6], **Elba Raimúndez**[1,2,6], **Lukas Refisch**[7,8], **Marcus Rosenblatt**[8,10], **Paul L. Stapor**[1,2], **Philipp Städter**[1,2], **Dantong Wang**[1,2], **Franz-Georg Wieland**[8,10], **Julio R. Banga**[5], **Jens Timmer**[8,10,14], **Alejandro F. Villaverde**[5], **Sven Sahle**[3], **Clemens Kreutz**[7,8,14], **Jan Hasenauer**[1,2,6‡]*, **Daniel Weindl**[1‡]

**1** Institute of Computational Biology, Helmholtz Zentrum München—German Research Center for Environmental Health, Neuherberg, Germany, **2** Center for Mathematics, Technische Universität München, Garching, Germany, **3** BioQUANT/COS, Heidelberg University, Heidelberg, Germany, **4** Department of Applied Mathematics II, University of Vigo, Vigo, Galicia, Spain, **5** BioProcess Engineering Group, IIM-CSIC, Vigo, Galicia, Spain, **6** Faculty of Mathematics and Natural Sciences, University of Bonn, Bonn, Germany, **7** Faculty of Medicine and Medical Center, Institute of Medical Biometry and Statistics, University of Freiburg, Freiburg, Germany, **8** Freiburg Center for Data Analysis and Modeling (FDM), University of Freiburg, Freiburg, Germany, **9** Department of Systems Biology, Harvard Medical School, Boston, Massachusetts, USA, **10** Institute of Physics, University of Freiburg, Freiburg, Germany, **11** Ragon Institute of MGH, MIT and Harvard, Cambridge, Massachusetts, USA, **12** Department of Biological Engineering, Massachusetts Institute of Technology, Cambridge, Massachusetts, USA, **13** Heidelberg Institute for Theoretical Studies (HITS gGmbH), Heidelberg, Germany, **14** Signalling Research Centres BIOSS and CIBSS, University of Freiburg, Freiburg, Germany

☾ These authors contributed equally to this work.
‡ JH and DW also contributed equally to this work.
* jan.hasenauer@uni-bonn.de

**Data Availability Statement:** The authors confirm that all data underlying the findings are fully available without restriction. Specifications of PEtab, the PEtab Python library, as well as links to

## Abstract

Reproducibility and reusability of the results of data-based modeling studies are essential. Yet, there has been—so far—no broadly supported format for the specification of parameter estimation problems in systems biology. Here, we introduce PEtab, a format which facilitates the specification of parameter estimation problems using Systems Biology Markup Language (SBML) models and a set of tab-separated value files describing the observation model and experimental data as well as parameters to be estimated. We already implemented PEtab support into eight well-established model simulation and parameter estimation toolboxes with hundreds of users in total. We provide a Python library for validation and modification of a PEtab problem and currently 20 example parameter estimation problems based on recent studies.

examples, and all supporting software tools are available at https://github.com/PEtab-dev/PEtab a snapshot is available at https://doi.org/10.5281/zenodo.3732958. All original content is available under permissive licenses.

**Funding:** This work was supported by the European Union's Horizon 2020 research and innovation program (CanPathPro; Grant no. 686282; J.H., D.W., P.L.S., E.D., S.M., A.F.V., J.R. B.), the German Federal Ministry of Education and Research (Grant no. 01ZX1916A; D.W., P.L. & 01ZX1705A; J.H., P.L. & Grant. no. 031L0159C; J. H. & de.NBI ModSim1, 031L0104A; F.T.B. & EA: Sys, 031L0080; J.E, L.R & Grant no. 031L0048; S. K., F.G.W. & 01ZX1310B; E.R.), the German Federal Ministry of Economic Affairs and Energy (Grant no. 16KN074236; D.P.), the German Research Foundation (Grant no. HA7376/1-1; Y.S.; CIBSS-EXC-2189-2100249960-390939984; C.K., J.T.; project ID: 272983813 - TRR 179; M.R. & Clusters of Excellence EXC 2047 and EXC 2151; E. R., J.H.), the Deutsche Krebshilfe (Grant no. 70112355; A.L.H.), the Human Frontier Science Program (Grant no. LT000259/2019-L1; F.F.), the National Institute of Health (Grant no. U54-CA225088; F.F.). The funders had no role in study design, data collection and analysis, decision to publish, or preparation of the manuscript.

**Competing interests:** The authors have declared that no competing interests exist.

## Author summary

Parameter estimation is a common and crucial task in modeling, as many models depend on unknown parameters which need to be inferred from data. There exist various tools for tasks like model development, model simulation, optimization, or uncertainty analysis, each with different capabilities and strengths. In order to be able to easily combine tools in an interoperable manner, but also to make results accessible and reusable for other researchers, it is valuable to define parameter estimation problems in a standardized form. Here, we introduce PEtab, a parameter estimation problem definition format which integrates with established systems biology standards for model and data specification. As the novel format is already supported by eight software tools with hundreds of users in total, we expect it to be of great use and impact in the community, both for modeling and algorithm development.

## Introduction

Dynamical modeling is central to systems biology, providing insights into the underlying mechanisms of complex phenomena [1]. It enables the integration of heterogeneous data, the testing and generation of hypotheses, and experimental design. However, to achieve this, the unknown model parameters commonly need to be inferred from experimental observations.

Various software tools exist for simulating models and inferring parameters [2–10], which implement various methods and algorithms. Many of these tools support community standards for model specification to facilitate reproducibility, interoperability and reusability. In particular the Systems Biology Markup Language (SBML) [11], CellML [12] and the BioNet-Gen Language (BNGL) [13] are widely used.

The Simulation Experiment Description Markup Language (SED-ML) builds on top of such model definitions and allows for a machine-readable description of simulation experiments based on XML [14]. Also more complex simulation experiments like parameter scans can be encoded, and a human-readable adaptation is provided by the phraSED-ML format [15]. Similarly, the XML-based Systems Biology Results Markup Language (SBRML) was designed to associate models with experimental data and share simulation experiment results in a machine-readable way [16]. Like SED-ML, SBRML can also be used for parameter scans. Complementary, SBtab is a set of table-based conventions for the definition of experimental data and models designed for human-readability and -writability [17].

However, parameter estimation is so far not in the scope of any of the available formats, and important information for it, like the definition of a noise model, is missing. Parameter estimation toolboxes usually use their own specific input formats, making it difficult for the user to switch between tools to benefit from their complementary functionalities and hindering reusability and reproducibility.

Based on our experience with parameter estimation and tool development for systems biology, we developed PEtab, a tabular format for specifying parameter estimation problems. This includes the specification of biological models, observation and noise models, experimental data and their mapping to the observation model, as well as parameters in an unambiguous way.

## Design and implementation

### Scope

The scope of PEtab is the full specification of parameter estimation problems in typical systems biology applications. In our experience, a typical setup of data-based modeling starts either with (i) the model of a biological system that is to be calibrated, or with (ii) experimental data that are to be integrated and analyzed using a computational model. Measurements are linked to the biological model by an observation and noise model. Often, measurements are taken after some perturbations have been applied, which are modeled as derivations from a generic model (Fig 1A). Therefore, one goal was to specify such a setup in the least redundant way. Furthermore, we wanted to establish an intuitive, modular, machine- and human-readable and -writable format that makes use of existing standards.

### A   Typical experimental and model setup and workflow

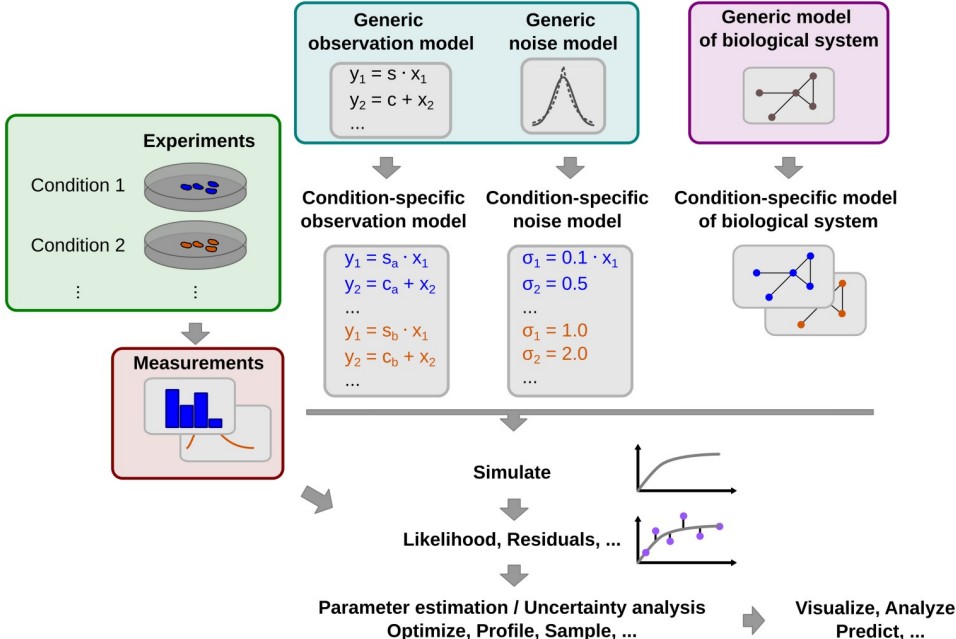

### B   Representation of the workflow elements in PEtab

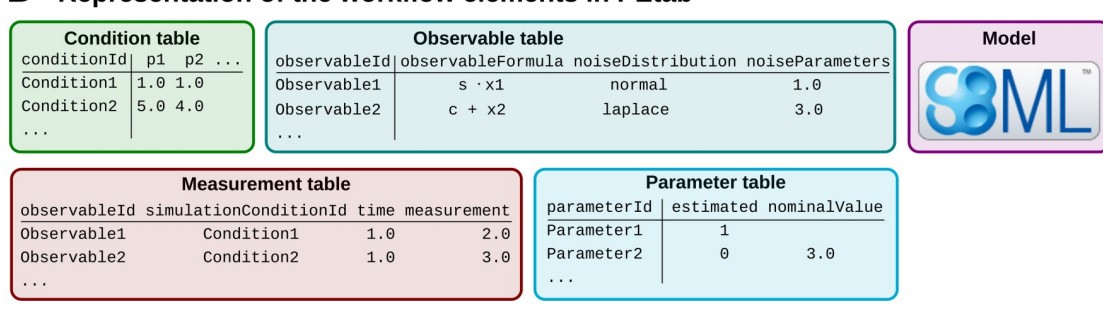

**Fig 1. Specifying parameter estimation problems in PEtab.** (A) Example of a typical setup for data-based modeling. Usually, a model of a biological system is developed and calibrated based on measurements from perturbation experiments, which are linked to the biological model by an observation model. Different instances of a generic model are used to account for different perturbations or measurement setups. (B) Simplified illustration of how different entities from (A) map to different PEtab files (not all table columns are shown).

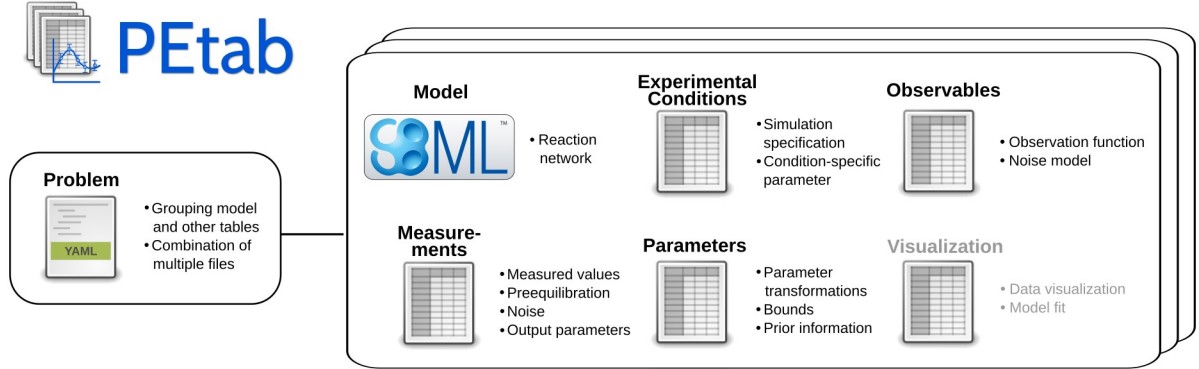

**Fig 2. Overview of PEtab files and the most important features.** PEtab consists of a model in the SBML format and several tab-separated value (TSV) files to specify measurements and link them to the model. A visualization file can be provided optionally. A YAML file can be used to group the aforementioned files unambiguously.

## PEtab problem specification format

PEtab defines parameter estimation problems using a set of files that are outlined in Fig 2. A detailed specification of PEtab version 1 is provided in supplementary file S1 File, as well as at https://github.com/PEtab-dev/PEtab. Additionally, we created a tutorial illustrating how to set up a PEtab problem, covering the most common features (supplementary file S2 File). Further example problems can be found at https://github.com/Benchmarking-Initiative/Benchmark-Models-PEtab. The different files specify the biological model, the observation model, experimental conditions, measurements, parameters and visualizations (Fig 1B). These files are described in more detail in the following.

**Model (SBML):** File specifying the biological process using the established and well-supported SBML format [11]. Any existing SBML model can be used without modification. All versions of SBML are supported by PEtab and can be used if the specific toolbox supports it.

**Experimental conditions (TSV):** File specifying the condition(s), such as drug stimuli or genetic backgrounds, under which the experimental data were collected. These experimental conditions specify model properties that are altered between conditions, and allow for a hierarchical specification of model properties (Fig 3A). If simulation conditions are used for pre-equilibration—meaning that some experiment started from the equilibrium reached for another condition—specific model states can be marked for re-initialization (Fig 3B).

**Observables (TSV):** File linking model properties such as state variables and parameter values to measurement data via observation functions and noise models. Various noise models including normal and Laplace distributions are supported, and noise model parameters can be estimated. Observables can be on linear or logarithmic scale.

**Measurements (TSV):** File specifying and linking experimental data to the experimental conditions and the observables via the respective identifiers. Optionally, simulation conditions for pre-equilibration can be defined (Fig 3B). Parameters that are relevant for the observation process of a given measurement, such as offsets or scaling parameters, can be provided along with the measured values. This allows for overriding generic output parameters in a measurement-specific manner (Fig 3A).

**Parameters (TSV):** File defining the parameters to be estimated, including lower and upper bounds as well as transformations (e.g., linear or logarithmic) to be used in parameter estimation. Furthermore, prior information on the parameters can be specified to inform starting points for parameter estimation, or to perform Bayesian inference.

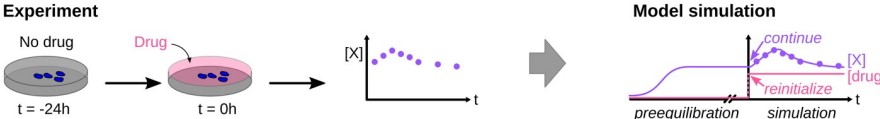

**A  Condition- / measurement-specific parameter overriding**

**1.** Generic model parameters / output parameters  **2.** Replicate for all simulation conditons  **3.** Measurement-specific output parameters  **4.** Condition-specific parameters  **5.** Nominal / estimated parameters

**B  Pre-equilibration / reinitialization**

**Fig 3. Parameter hierarchy and pre-equilibration in PEtab.** (A) Illustration of possibilities and precedence of parameter overriding at different stages. The generic model parameter vector, as specified in the SBML model, can be overridden via the observable, measurement, condition and parameter tables, differentially for conditions and measurement points to account for different model inputs or observational model parameters. The parameters that are overridden in each step are indicated with thicker cell borders. Individual parameters can be set to specific values or marked to be estimated (as here p1). (B) In an often encountered experimental setup, a biological system is under some "baseline" condition and assumed to be in equilibrium (e.g., here depicted for after 24h incubation) before a perturbation is applied. If the equilibrium state of the system is not known a priori, such a setup can be modeled by simulating the system until an apparent steady-state is reached (pre-equilibration). To simulate the perturbation, a subset of model states are reinitialized.

**Visualization (TSV):** Optional visualization file specifying how to combine data and simulations for plotting. Different plots such as time-course or dose-response curves can be automatically created based on this file using the PEtab Python library described below. This allows, for example, to quickly create visualizations to inspect parameter estimation results. A default visualization file can be automatically generated.

**PEtab problem file (YAML):** File linking all of the above-mentioned PEtab files together. This allows combinations of, e.g., multiple models or measurement files into a single parameter estimation problem, as well as easy reuse of various files in different parameter estimation problems (e.g., for model selection). The current YAML version 1.2 is used here.

We designed PEtab to cover common features needed for parameter estimation. The TSV files comprise different mandatory columns. These provide all necessary information to define an objective function like the $\chi^2$ or likelihood function. However, some methods tailored to specific problems require additional information to estimate the unknown parameters. To acknowledge this, we allow for optional application-specific extensions in addition to the required columns in the PEtab files, e.g., if some parameters can be calculated analytically using hierarchical optimization approaches [18].

## PEtab library

To facilitate easy usability, PEtab (https://github.com/PEtab-dev/PEtab) comes with detailed documentation describing the specific format of each of the different files in a concise yet comprehensive manner. Additionally, we provide a Python-based library that can be used to read, modify, write, and validate existing PEtab problems. Furthermore, the PEtab library provides functionality to package PEtab files into COMBINE archives [19]. After parameter estimation, the modeler usually investigates how well the model fits the experimental data. To support this, the PEtab library provides various visualization routines to analyze data and parameter estimation results.

## Results

### PEtab support in established tools

We implemented support for PEtab in currently eight systems biology toolboxes, namely COPASI [2], AMICI [6], pyPESTO [20], pyABC [21], Data2Dynamics [5], dMod [10], parPE [18], and MEIGO [4]. These toolboxes provide a broad range of distinct features for model creation, model simulation, parameter inference, and uncertainty quantification (Table 1). Combining different tools with complementary features is often desirable. However, in practice this was hitherto hampered by the substantial overhead of tedious and error-prone re-implementation of the parameter estimation problem in the specific format required by the respective tool. With all of these tools now supporting PEtab, a user can more easily combine different tools and make use of their specific strengths. For example, one can use COPASI for model creation and testing, AMICI for efficient simulation of large models, pyPESTO for multi-start local optimization and sampling, or MEIGO for global scatter searches, and Data2Dynamics or

**Table 1. Non-exhaustive overview of the functionality offered by the different toolboxes currently supporting the PEtab format.** The list of supporting tools and functionality covered by the respective tools may increase over time. An updated overview is available on the PEtab website. Darker colors indicate more accurate, scalable, or broader functionality compared to basic implementations.

| | COPASI | D2D | dMod | MEIGO | AMICI | parPE | pyABC | pyPESTO |
|---|---|---|---|---|---|---|---|---|
| **Interface / Language** | Graphical interface | MATLAB | R | MATLAB | C++, Python, MATLAB | C++ | Python | Python |
| **Model construction** | Advanced | Basic | Basic | No | No | No | No | No |
| **Model simulation** | Accurate | Accurate, Scalable | Accurate, Scalable | Uses AMICI | Accurate, Scalable | Uses AMICI | Uses AMICI | Uses AMICI |
| **Gradient computation** | Approximative | Accurate | Accurate | Uses AMICI | Accurate, Scalable | Uses AMICI | No | Uses AMICI |
| **Gradient-free (global) parameter estimation** | Multiple algorithms | Basic | No | Metaheuristic algorithms | No | No | No | Basic |
| **Gradient-based parameter estimation** | Basic | Multiple local optimizers | Multiple local optimizers | Metaheuristic algorithms | No | Multiple local optimizers | No | Multiple local optimizers |
| **Parameter profile likelihood** | Basic | Advanced | Advanced | No | No | No | No | Advanced |
| **Prediction profile likelihood** | Basic | Advanced | Advanced | No | No | No | No | No |
| **Parameter sampling** | No | Basic | No | No | No | No | Adaptive SMC algorithms | Multiple MCMC algorithms |
| **Simulation of stochastic models** | Multiple algorithms | No | No | No | No | No | No | No |
| **Parameter inference for stochastic models** | Basic | No | No | Basic | No | No | Scalable | Basic |
| **Particular strengths** | Advanced modeling Strong allrounder Graphical interface | Powerful gradient based optimization Advanced profiling Strong allrounder | Powerful gradient based optimization Strong allrounder | Powerful metaheuristic optimization | Highly scalable simulation & gradient comp. | Highly scalable optimization for large-scale models | Scalable likelihood-free inference | Allrounder Multiple MCMC samping methods |

**Table 2. Overview of supported PEtab features in different tools, based on passed test cases of the PEtab test suite.** The first character indicates whether computing simulated data is supported and simulations are correct (✓) or not (-). The second character indicates whether computing $\chi^2$ values of residuals are supported and correct (✓) or not (-). The third character indicates whether computing likelihoods is supported and correct (✓) or not (-). An up-to-date overview of supported features is maintained on the PEtab GitHub page.

| Test-case | AMICI | Copasi | D2D | dMod | MEIGO | parPE | pyABC | pyPESTO |
|---|---|---|---|---|---|---|---|---|
| Basic simulation | ✓✓✓ | ✓- - | ✓✓✓ | ✓✓✓ | ✓✓✓ | - -✓ | ✓✓✓ | ✓✓✓ |
| Multiple simulation conditions | ✓✓✓ | ✓- - | ✓✓✓ | ✓✓✓ | ✓✓✓ | - -✓ | ✓✓✓ | ✓✓✓ |
| Numeric initial compartment sizes in condition table | - - - | ✓- - | ✓✓✓ | ✓✓✓ | ✓✓✓ | - - - | - - - | - - - |
| Numeric initial concentration in condition table | ✓✓✓ | ✓- - | ✓✓✓ | ✓✓✓ | ✓✓✓ | - -✓ | ✓✓✓ | ✓✓✓ |
| Numeric noise parameter overrides in measurement table | ✓✓✓ | ✓- - | ✓✓✓ | ✓✓✓ | ✓✓✓ | - -✓ | ✓✓✓ | ✓✓✓ |
| Numeric observable parameter overrides in measurement table | ✓✓✓ | ✓- - | ✓✓✓ | ✓✓✓ | ✓✓✓ | - -✓ | ✓✓✓ | ✓✓✓ |
| Observable transformations to log scale | ✓-✓ | ✓- - | ✓✓✓ | ✓✓- | ✓✓✓ | - -✓ | ✓-✓ | ✓-✓ |
| Observable transformations to log10 scale | ✓-✓ | ✓- - | ✓✓✓ | ✓✓- | ✓✓✓ | - -✓ | ✓-✓ | ✓-✓ |
| Parametric initial concentrations in condition table | ✓✓✓ | ✓- - | ✓✓✓ | ✓✓✓ | ✓✓✓ | - -✓ | ✓✓✓ | ✓✓✓ |
| Parametric noise parameter overrides in measurement table | ✓✓✓ | ✓- - | ✓✓✓ | ✓✓✓ | ✓✓✓ | - -✓ | ✓✓✓ | ✓✓✓ |
| Parametric observable parameteroverrides in measurement table | ✓✓✓ | ✓- - | ✓✓✓ | ✓✓✓ | ✓✓✓ | - -✓ | ✓✓✓ | ✓✓✓ |
| Parametric overrides in condition table | ✓✓✓ | ✓- - | ✓✓✓ | ✓✓✓ | ✓✓✓ | - -✓ | ✓✓✓ | ✓✓✓ |
| Partial pre-equilibration | ✓✓✓ | - - - | ✓✓✓ | ✓✓✓ | ✓✓✓ | - -✓ | ✓✓✓ | ✓✓✓ |
| Pre-equilibration | ✓✓✓ | ✓- - | ✓✓✓ | ✓✓✓ | ✓✓✓ | - -✓ | ✓✓✓ | ✓✓✓ |
| Replicate measurements | ✓✓✓ | ✓- - | ✓✓✓ | ✓✓✓ | ✓✓✓ | - -✓ | ✓✓✓ | ✓✓✓ |
| Time-point specific overrides in the measurement table | - - - | - - - | ✓✓✓ | ✓✓✓ | ✓✓✓ | - - - | - - - | - - - |

dMod for profiling. The ease of switching between tools also provides the opportunity to easily reproduce and verify results, e.g., whether different tools yield similar results. Additionally, developers can compare the performance of newly developed methods with existing algorithms implemented in different toolboxes, independent of the programming language, to select the most appropriate one for a given setting.

## PEtab test suite and examples

Along with introducing PEtab support to different tools, we have set up a test suite with various toy problems and reference values that can be used by other tool developers to assess and verify PEtab support in their software packages. The specific status of the PEtab support of the different tools is provided in Table 2 and continuously updated on the PEtab GitHub webpage. The test cases are based on SBML level 2 version 4 which is supported by all considered toolboxes.

To demonstrate the various features and the broad applicability of PEtab, we provide a growing collection of currently 20 example parameter estimation problems in the PEtab format largely based on a previously published benchmark collection [22]. These models can be used as templates for creating new PEtab problems and for method development and testing.

## Availability and future directions

PEtab complements existing standards for model definition by facilitating the specification of complex estimation problems using tabular text files, defining experimental measurements and linking model entities and measurements via observables and a noise model.

The specification of the PEtab format, the PEtab Python library, as well as links to examples, a web-based validation tool, and all supporting software are available at https://github.com/ PEtab-dev/PEtab. A snapshot is available at https://doi.org/10.5281/zenodo.3732958. PEtab and all original content presented here is available under permissive licences. For any

questions or requests related to PEtab, we encourage interested users to approach us via the Issues function in the aforementioned GitHub repository, or the respective tool repositories for more specific queries.

We developed PEtab to cover the most common features needed for parameter estimation in the context of dynamic modeling. However, as multiple model formats as well as a multitude of tailored parameter estimation methods exist, which require different information, we could not cover every aspect. While at the time of writing, PEtab only allows for models defined in the SBML format, the PEtab format is general enough to be integrated with other model specification formats like CellML and rule-based formats [13] in the future. Additionally, other formats like SBtab [17] or Antimony [23] provide converters to SBML and can therefore also indirectly be used together with PEtab. Recently, new methods have been developed to estimate parameters in a hierarchical manner [18], including from qualitative data [24, 25]. PEtab could be extended to also allow for these types of measurements. To cover the most important needs, we invite users and developers to suggest new features to be supported by PEtab. We formed a maintainer team comprising developers of all supporting toolboxes to facilitate long-term support and improvement of PEtab. We encourage additional toolbox developers to implement support for PEtab. As an example, since the preprint publication of this manuscript, PEtab has already been adopted as the input format for a newly developed tool, SBML2Julia [26].

As PEtab is already supported by software tools with hundreds of users in total, we envisage that it will facilitate reusability, reproducibility and interoperability. We expect that a common specification format will prove helpful for users as well as developers of parameter estimation tools and methods in systems biology.

## Supporting information

**S1 File. PEtab specification.** Detailed format description of PEtab version 1.
(PDF)

**S2 File. PEtab tutorial.** Step-by-step instructions for creating PEtab files for an application example.
(PDF)

## Acknowledgments

We thank Dagmar Waltemath for helpful discussions.

## Author Contributions

**Conceptualization:** Leonard Schmiester, Yannik Schälte, Jan Hasenauer, Daniel Weindl.

**Methodology:** Leonard Schmiester, Yannik Schälte, Frank T. Bergmann, Fabian Fröhlich, Jan Hasenauer, Daniel Weindl.

**Software:** Leonard Schmiester, Yannik Schälte, Frank T. Bergmann, Tacio Camba, Erika Dudkin, Janine Egert, Fabian Fröhlich, Lara Fuhrmann, Adrian L. Hauber, Svenja Kemmer, Polina Lakrisenko, Carolin Loos, Simon Merkt, Wolfgang Müller, Dilan Pathirana, Elba Raimúndez, Lukas Refisch, Marcus Rosenblatt, Paul L. Stapor, Philipp Städter, Dantong Wang, Franz-Georg Wieland, Julio R. Banga, Jens Timmer, Alejandro F. Villaverde, Sven Sahle, Clemens Kreutz, Jan Hasenauer, Daniel Weindl.

**Supervision:** Wolfgang Müller, Julio R. Banga, Jens Timmer, Alejandro F. Villaverde, Clemens Kreutz, Jan Hasenauer.

**Writing – original draft:** Leonard Schmiester, Yannik Schälte, Jan Hasenauer, Daniel Weindl.

**Writing – review & editing:** Leonard Schmiester, Yannik Schälte, Frank T. Bergmann, Tacio Camba, Erika Dudkin, Janine Egert, Fabian Fröhlich, Lara Fuhrmann, Adrian L. Hauber, Svenja Kemmer, Polina Lakrisenko, Carolin Loos, Simon Merkt, Wolfgang Müller, Dilan Pathirana, Elba Raimúndez, Lukas Refisch, Marcus Rosenblatt, Paul L. Stapor, Philipp Städter, Dantong Wang, Franz-Georg Wieland, Julio R. Banga, Jens Timmer, Alejandro F. Villaverde, Sven Sahle, Clemens Kreutz, Jan Hasenauer, Daniel Weindl.

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
