## [Decision Letter · Decision Letter 0]

23 Oct 2020

Dear Mr. Schälte,

Thank you very much for submitting your manuscript "PEtab -- Interoperable Specification of Parameter Estimation Problems in Systems Biology" for consideration at PLOS Computational Biology. As with all papers reviewed by the journal, your manuscript was reviewed by members of the editorial board and by several independent reviewers. The reviewers appreciated the attention to an important topic. Based on the reviews, we are likely to accept this manuscript for publication, providing that you modify the manuscript according to the review recommendations.

The reviewers were positive about the PEtab format, however it will be useful to include in the manuscript examples, as suggested by reviewer 1. A basic and more advanced example of PEtab can help potential users.

Sincerely,

Dina Schneidman

Software Editor

PLOS Computational Biology

[LINK]

Reviewer's Responses to Questions

**Comments to the Authors:**

Reviewer #1: This manuscript is a report about a new standardized data format called PEtab. The data represented in the PEtab format is that needed to solve curve fitting (or model parameterization) problems. Examples of PEtab files (or rather collections of files) are provided at the GitHub repository cited in the manuscript. The format is supported by various software tools developed to support systems biology modeling. Importantly, these tools include useful ones, and moreover, the collection offers an array of non-overlapping capabilities. The description of PEtab is probably sufficiently detailed for those who may wish to adopt the format (as a user or developer of a new or existing software package). PEtab is potentially useful in that it could make it easier for modelers to leverage multiple software packages, because problem setup should be at least similar now for all of the tools supporting PEtab. To my best knowledge, there is no standardized format for the definition of curve-fitting problems in the systems biology modeling field.

A concern is potential impact. The authors have not provided a compelling demonstration of the benefit of having a standardized format. Time will tell if the format is adopted and proves to be helpful in moving the field of systems biology modeling forward.

Another concern is that the manuscript lacks an accessible introduction to PEtab for the new user. It would be nice if the authors could add a step-by-step problem-setup guide that walks the reader through setup of one or more simple fitting problems using PEtab from start to finish. I think this addition to the text would greatly improve accessibility of the format. The guidance provided in the manuscript at present is of a general nature, and problem setup seems to be somewhat complicated. Consideration of a specific demonstration problem (or a few of different kinds, ideally) would make the guidance on using the new format more concrete.

Reviewer #2: Summary of research and overall impression:

PEtab is a data format for specifying systems biology parameter estimation problems based on Tab-Separated Values (TSV) files. Currently, PEtab only allows for models defined in SBML format and is supported by eight systems biology toolboxes, with long term support facilitated by a maintainer team comprising developers from these toolboxes.

Parameter estimation problems are specified by an SBML model specifying the biological process under investigation and the following TSV files:

a. A measurement file to fit the model to

b. A condition file specifying model inputs and condition-specific parameters

c. An observable file specifying the observation model

d. A parameter file specifying optimization parameters and related information

e. (optional) A simulation file, which has the same format as the measurement file, but contains model simulations

f. (optional) A visualization file, which contains specifications how the data and/or simulations should be plotted by the visualization routines

Additionally a PEtab problem file, written in YAML format (currently 1.2), is needed to link all of the afore-mentioned PEtab files.

Links to Github repositories with detailed documentations describing the formats of each file as well as installation instructions for the python library developed to create, check, visualize and work with PEtab files are provided in the manuscript. A wide range of system biology parameter estimation PEtab examples are also provided as benchmark problems.

The authors acknowledge the existing tools which are used for simulating models and inferring parameters and highlight the community standards used by these tools. The authors do well to highlight the importance of their tool by describing the lack of interoperability between current parameter estimation tools and the community based standards.

PEtab represents an excellent attempt to standardize parameter estimation problems in Systems Biology and will greatly improve the reusability, reproducibility and interoperability of such models in this domain. However further integration with other model- and rule-based formats is needed, as although inter-standard converters exist and can potentially convert these formats into SBML, they are known to be prone to errors and are not smoothly implemented.

Minor Comments

PEtab problem specification format:

“A detailed specification of PEtab version 1 is provided in supplementary file S1 File, as well as at https://github.com/PEtab-dev/PEtab. Further example problems can be found at https://github.com/Benchmarking-Initiative/Benchmark-Models-PEtab. The different files specify the biological model, the observation model, experimental conditions, measurements, parameters and visualizations (Fig 1B).”

The supplementary file and links to the Github repository for the source code and bench-marking implementations are most welcomed. I was able to run the bench-marking models on multiple supported platforms, further confirming the interoperability of the data format developed.

However I suggest adding a figure similar to Fig1B in the manuscript to the Github documentation page for further clarification and representation of the workflow elements in PEtab.

"Parameter hierarchy and pre-equilibrium in PEtab Fig 3 A and B":

The caption and graphical representation in Figure 3A needs further clarity. I believe that the authors are describing the ability to override parameters through the use of measurements or conditions accounting for observational parameters or model inputs but this is not entirely clear from the figures.

"We formed a maintainer team comprising developers of all supporting toolboxes to facilitate long-term support and improvement of PEtab."

This is an excellent initiative by the authors and tool developers. Perhaps a general point of contact to disseminate queries to the specific tool developers should be mentioned. Perhaps through the github page or similar.

**Have all data underlying the figures and results presented in the manuscript been provided?**

Reviewer #1: Yes

Reviewer #2: None

PLOS authors have the option to publish the peer review history of their article (what does this mean?). If published, this will include your full peer review and any attached files.

Reviewer #1: No

Reviewer #2: No
---

## [Editor Report · Decision Letter 1]

18 Dec 2020

Dear Mr. Schälte,

We are pleased to inform you that your manuscript 'PEtab -- Interoperable Specification of Parameter Estimation Problems in Systems Biology' has been provisionally accepted for publication in PLOS Computational Biology.

Best regards,

Dina Schneidman

Software Editor

PLOS Computational Biology

---

## [Editor Report · Acceptance letter]

21 Jan 2021

PCOMPBIOL-D-20-01416R1 

PEtab -- Interoperable Specification of Parameter Estimation Problems in Systems Biology

Dear Dr Hasenauer,

I am pleased to inform you that your manuscript has been formally accepted for publication in PLOS Computational Biology. Your manuscript is now with our production department and you will be notified of the publication date in due course.

With kind regards,

Alice Ellingham
